# Optimization of Roasting Parameters for Recovery of Vanadium and Tungsten from Spent SCR Catalyst with Composite Roasting

**Bo Wang * and Qiaowen Yang**

School of Chemical and Environmental Engineering, China University of Mining and Technology-Beijing, Beijing 100083, China; abcyqw@sina.com
* Correspondence: xsywb2020@126.com

**Abstract:** Every year, large amounts of selective catalytic reduction (SCR) catalysts with losing catalytic activity and failing to be regenerated need to be regenerated, which will result in acute pollution. Recycling valuable metals from spent SCR catalysts can not only solve environmental problems, but also save resources. The process of sodium roasting and water leaching is able to effectively extract vanadium (V) and tungsten (W) from spent SCR catalysts. To improve the efficiencies of V and W, different sodium additives were first investigated in the roasting process. The results revealed that the process of NaCl-NaOH composite roasting and water leaching showed superior leaching efficiencies of V and W, which can reach 91.39% and 98.26%, respectively, and simultaneously, it can be found that adding low melting point NaOH promoted mass transfer as compared with the melting points of different sodium additives. Next, a single-factor experiment was conducted to investigate different roasting conditions, such as roasting temperature, roasting time, mass ratio of sodium additive and catalyst, and mass ratio of NaCl and NaOH, on the leaching efficiencies of V and W. Then, a three-level and four-factor orthogonal experiment and a weight matrix analysis were used to optimize the roasting parameters. The results showed that roasting temperature had the most significant effect on the leaching efficiencies of V and W, and the optimal roasting conditions were as follows: the roasting temperature was 750 °C, the roasting time was 2.5 h, the mass ratio of sodium additive and catalyst was 2.5, and the mass ratio of NaCl and NaOH was 1.5. Under the optimal roasting conditions, the leaching efficiencies of V and W were 93.25% and 99.17%, respectively. The results of XRD analysis inferred that $VO_2$ coming from the decomposition of $VOSO_4$ in spent SCR catalysts may first oxidize into $V_2O_5$ and then react with sodium additives to produce $NaVO_3$. The formation of titanium-vanadium oxide $((Ti_{0.5}V_{0.5})_2O_3)$ was a part reason of hindering the leaching of vanadium. With the increase of roasting temperature, $TiO_2$ converted into $Na_2Ti_3O_7$, which indicated that the main structure of the catalyst was destroyed, and simultaneously, more characteristic peaks of sodium metavanadate and sodium tungstate appeared, thus enhancing the leaching of V and W. Finally, it can be seen that the process of NaCl-NaOH roasting and water leaching remained higher leaching efficiencies of V and W and lower roasting temperature by comparing with leaching efficiencies of V and W in different processes of recycling SCR catalyst. The process of NaCl-NaOH composite roasting and water leaching provided a strategy with a highly efficient and clean route to leach V and W from spent SCR catalyst. The orthogonal experiment and weight matrix analysis in our study can be used as a reference to optimize the reaction conditions of a multiple indexes experiment.

**Keywords:** optimization of composite roasting parameters; recovery of spent SCR catalyst; vanadium; tungsten; orthogonal experiment; weight matrix analysis



## 1. Introduction

Energy is the driving force to promote the rapid development of economy; thus, efficient and clean utilization of existing resources is an important element for China's

economic take-off. Recently, different kinds of energies have been utilized in China, such as wind energy, solar energy, petroleum, natural gas, and coal. Among them, petroleum, natural gas, and coal have been widely employed in modern industry. It is reported that the coal reserves in China are about 1.0025 billion tons, ranked at number three in the world, while the petroleum reserves and natural gas reserves in China are approximately 104.01 million tons and 471 million cubic meters, ranked at number twelve and number ten in the world, respectively [1]. This shows that the energy structure in China presents "richness in coal, less gas and lack of oil". Therefore, with the increasing depletion of China's oil resources, coal plays an important role in energy structure and energy consumption [2]. However, in the process of coal mining, and utilization, especially in the coal combustion of coal-fired power plants, a series of environmental issues, such as acid rain and atmosphere pollution, are exposed [3].

The main pollutants in the flue gas emitted by coal-fired power plants are nitrogen oxides, sulfur oxides, and inhalable particles [4]. Among them, sulfur oxides and inhalable particles have been effectively controlled, and nitrogen oxides have been the first pollutant in flue gas [5]. The removal methods of nitrogen oxides in flue gas mainly includes selective catalytic reduction (SCR) technology and selective non-catalytic reduction (SNCR) technology. Compared with SNCR technology, SCR technology has proved as the more effective process to remove nitrogen oxides in the flue gas environment [6]. The SCR catalysts play an important role in the denitrification process, among which $V_2O_5$-$WO_3$($MoO_3$)/$TiO_2$ catalyst performs a superior catalytic activity, excellent mechanical strength, and perfect anti-inactivation ability [7–9]. However, the SCR catalysts have a definite lifetime, which is generally 3–4 years, resulting in being discarded after they lose catalytic activity and fail to be regenerated [10,11]. It is reported that in China, nearly 38,000 tons spent SCR catalyst will be produced every year [12]. Burying the spent SCR catalyst underground as hazardous waste will lead to serious environmental pollution [13]. At the same time, vanadium and tungsten as rare metals which can be recovered and applied in chemical industry as secondary energy [14]. Therefore, recycling SCR catalysts can not only solve the pollution problem of hazardous waste, but also produce tremendous economic benefits [15].

In recent years, different kinds of approaches have been reported for recycling V, W, and Ti from spent SCR catalysts. The recycling routes are summarized as the dry-wet combination method and the wet method by solid-solid reaction or liquid-solid reaction [16–20]. For the wet method, NaOH is often employed to leach spent catalysts, which is then followed by the development of some new processes, such as leaching catalysts by $Na_2CO_3$ solution or ammonia solution, pressure leaching, and electrochemistry leaching [21–24]. In addition, in order to increase the recovery efficiency of vanadium, sulfuric acid has been used to leach spent SCR catalysts, followed by the development of other approaches, such as hydrochloric acid leaching, oxalic acid leaching, and reducing acid leaching [25,26]. In the dry wet combination method, soda roasting and water leaching is one of the most effective and traditional routes to extract vanadium and tungsten from spent SCR catalyst, followed by the development of sodium hydroxide roasting and water leaching, sodium hydroxide roasting and acid leaching, and calcium roasting and acid leaching [27–30].

The leaching process is the most important step for recycling V and W from spent SCR catalysts. Elevating the leaching efficiencies of metals is helpful to increase the recovery efficiencies of metals and decrease the cost. In many approaches of recycling spent SCR catalyst, sodium roasting and water leaching has been proved to reach the highest leaching efficiencies of vanadium and tungsten, and reduce the discharge of wastewater at the same time, while acid methods or alkali methods failed to simultaneously elevate the leaching efficiencies of V and W and generate large amount of acidic or alkaline effluents [31]. Some sodium compounds, such as sodium hydroxide, sodium carbonate, sodium sulphate, and sodium chloride, are the common additives for sodium roasting. For some pure sodium additives, the leaching efficiencies of vanadium and tungsten often cannot reach higher values at the same time. For instances, Zhou et al. mixed 30 wt.% $Na_2CO_3$ with 10 g spent SCR catalyst in the crucible, roasted at 900 °C, and then leached V and W under the

conditions of 40 °C, a liquid–solid ratio of 3:1, for 3.0 h; however, the leaching efficiency of V was only 49.05% and the leaching efficiency of W was 99.02% [32]. Therefore, in our work, we proposed a process of composite sodium additives to react with spent SCR catalyst by roasting at high temperature. The routes can simultaneously improve the leaching efficiencies of vanadium and tungsten under lower reaction temperature.

## 2. Materials and Methods

### 2.1. Materials

The common SCR catalyst included $V_2O_5$-$WO_3$/$TiO_2$ and $V_2O_5$-$MoO_3$/$TiO_2$. Among them, the $V_2O_5$-$MoO_3$/$TiO_2$ catalyst performed superior arsenic resistance, and simultaneously, showed the poor mechanical strength, which meant that it was difficult to be machined. Therefore, the $V_2O_5$-$WO_3$/$TiO_2$ catalyst was mainly applied in the flue gas denitrification process. The shapes of SCR catalyst can be divided into honeycomb-type, plate-type, and corrugated plate-type, among which the honeycomb-type SCR catalyst is the most widely employed in the industry because the phenomenon of accumulating dust, which blocks the channels, is less likely to occur [33]. In our study, the spent SCR catalyst of honeycomb-type $V_2O_5$-$WO_3$/$TiO_2$ was used to react with composite sodium additives by roasting. A spent SCR catalyst meant that the $NO_x$ conversion efficiency of the SCR catalyst was less than 50% [34]. The spent SCR catalysts were obtained from a coal-fired power plant in Shanghai, China. In order to minimize errors as much as possible and keep uniformity, we utilized the spent SCR catalyst collected from the same batch during the experiment. The morphology of spent SCR catalysts is shown in Figure 1.

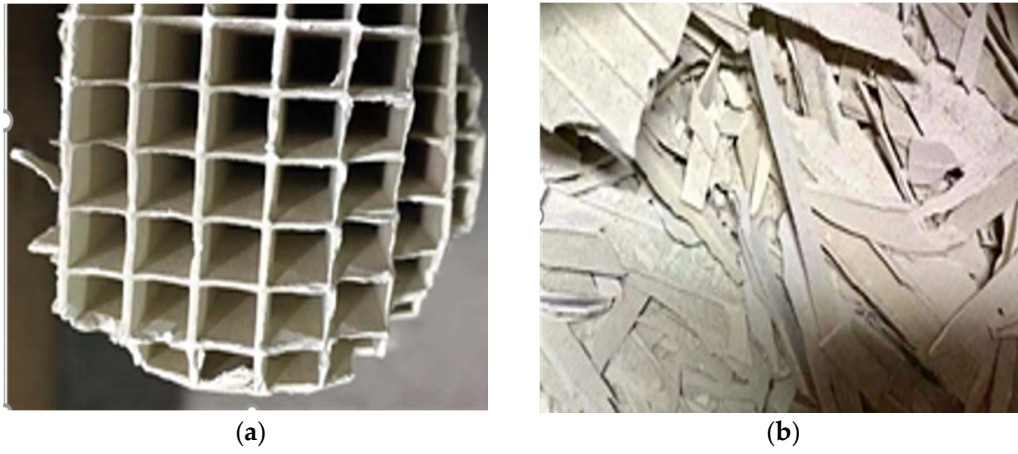

| (a) | (b) |

**Figure 1.** The macroscopic morphology of spent SCR catalysts (**a**) before crushing and (**b**) after crushing.

All of the chemical reagents employed in our study were analytical grade and were purchased from Aladdin reagent (Shanghai) Co., Ltd., Shanghai, China.

### 2.2. Characteristics

Chemical composition analysis of the catalysts was performed by X-ray diffraction and X-ray fluorescence spectrum (XRF, SPECTRO XEPOS, SPECTRO Analytical Instruments Company, Germany). The mineral phases of roasted clinker were identified by X-ray diffraction (XRD, D8 ADVANCE, Bruker Company, Karlsruhe, Germany). The concentration of vanadium and tungsten in the leaching solution was determined by UV-VIS spectrophotometry (UV 723E, 325-1000 nm, Shanghai Hongji Instrument Equipment Company, Shanghai, China).

### 2.3. Experimental Procedure

To remove fly ash and soluble impurities on the surface, the spent SCR catalysts were pretreated by being leached in hot water for 1.0 h and filtered. The filter residue was dried at 110 °C for 4.0 h. Then, in order to increase contact area among matters, it was fully ground and sieved at a size of 60 mesh. The treated spent SCR catalyst and sodium additives were evenly mixed in a nickel crucible based on appropriate proportions, followed by roasting at 450–900 °C for 0.5–3.0 h, after which the leaching of roasted clinker was performed in a 900 mL glass beaker with deionized water at a liquid–solid ratio of 8 mL/g and 90 °C for 2.0 h, simultaneously, keeping the stirring speed at 200 r/min. When the leaching solution was cooled to room temperature, the solution contained vanadium and tungsten, and titanium slag was finally obtained by filtering.

The filtrate was collected and the concentration of the leached elements was examined using an ultraviolet spectrophotometer. The concentration of vanadium and tungsten in the leaching solution was determined at the wavelength of 420 nm and 430 nm by an ultraviolet spectrophotometer. The leaching efficiency can be expressed with the following formula:

$$\eta = \frac{V \cdot C_i}{m \cdot W_i} \times 100\% \tag{1}$$

where $\eta$ is the leaching efficiency of vanadium or tungsten in the leaching solution, %; $C_i$ is the concentration of vanadium or tungsten in the leaching solution, g/mL; V is the volume of the leaching solution, mL; m is the mass of added spent SCR catalyst, g; $W_i$ is the content of vanadium or tungsten in the added spent SCR catalyst, wt.%.

In the single-factor experiment, we changed one of the roasting conditions that included roasting temperature, roasting time, m(additives)/m(catalyst), and m(NaCl)/m(NaOH), and kept the other roasting conditions unchanged. According to the results of the single-factor experiment, three levels of higher leaching efficiencies of vanadium and tungsten were selected from each roasting factor; thus, a three-level and four-factor orthogonal test was designed to optimize the roasting conditions. Because the orthogonal test had two response values (leaching efficiencies of V and W), the weight matrix method was utilized to acquire the optimal reaction conditions by comparing the weights of each response values. In our study, the leaching efficiency of V was as important as the leaching efficiency of W, so that the total matrix was equal to the mean value of the weight matrix of the orthogonal test results for leaching V and W. Finally, we compared the leaching efficiencies of V and W for different processes of recycling spent SCR catalysts.

## 3. Results and Discussion

### 3.1. Feedstock Composition

Figure 2 shows the XRD spectra of spent SCR catalysts. It was observed that only the characteristic peaks of anatase phase $TiO_2$ appeared, which meant that anatase phase $TiO_2$ were the main structure in the spent SCR catalyst. The characteristic peaks of other oxides, such as $V_2O_5$, $WO_3$, $SiO_2$, and $Al_2O_3$, failed to show in Figure 2, which was attributed to the lower content and the uniform distribution for these matters on the $TiO_2$ carrier. The SCR catalysts were analyzed by X-ray fluorescence spectrum (XRF). The chemical composition of spent SCR catalysts is presented in Table 1. As shown in Table 1, because $TiO_2$ was used as the carrier in the SCR catalyst, which played a role in dispersing active components, the content of $TiO_2$ was generally as high as 81.46 wt.%. $V_2O_5$ was the active component, and as long as a little $V_2O_5$ was added, the catalyst can perform superior catalytic activity; thus, the content of $V_2O_5$ was only 0.82 wt.%. To further improve the catalytic activity, 4.75 wt.% of $WO_3$ was added into the catalyst. In order to enhance the mechanical strength and poisoning resistance, silicon dioxide ($SiO_2$), aluminum oxide ($Al_2O_3$), and calcium oxide (CaO), which can add up to 11.16 wt.%, were also added to the catalyst. It was noticed that the spent SCR catalysts also contained a sulfur element, which meant some sulfates may exist in spent SCR catalysts. Zhang et al. thought that the sulfates in spent SCR catalyst may be $VOSO_4$ and $CaSO_4$ [26]. Based on the feedstock composition

of the spent SCR catalysts, it can be inferred that the following possible reactions in the process of NaOH-NaCl composite roasting are present:

$$4NaCl + O_2 = 2Na_2O + 2Cl_2 \tag{2}$$

$$Na_2O + V_2O_5 = 2NaVO_3 \tag{3}$$

$$Cl_2 + VOSO_4 + Na_2O = 2NaCl + NaVO_3 + SO_3 \tag{4}$$

$$2NaOH + V_2O_5 = 2NaVO_3 + H_2O \tag{5}$$

$$Na_2O + WO_3 = Na_2WO_4 \tag{6}$$

$$2NaOH + WO_3 = Na_2WO_4 + H_2O \tag{7}$$

$$2xTiO_2 + (2y - 4x)Na_2O = 2Na_{(2y-4x)}Ti_xO_y \tag{8}$$

$$xTiO_2 + (2y - 4x)NaOH = Na_{(2y-4x)}Ti_xO_y + (y - 2x)H_2O \tag{9}$$

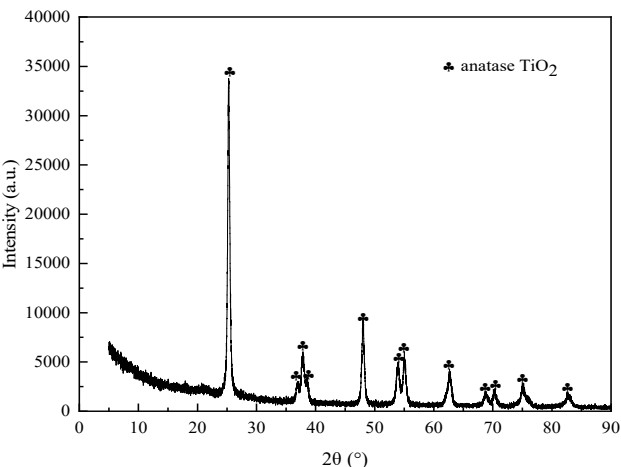

**Figure 2.** The XRD spectra of spent SCR catalysts.

**Table 1.** Chemical composition of fresh and spent SCR catalysts.

| Composition | TiO$_2$ | SiO$_2$ | WO$_3$ | CaO | Al$_2$O$_3$ | V$_2$O$_5$ | SO$_3$ | Others |
|---|---|---|---|---|---|---|---|---|
| Content (wt.%) | 81.46 | 7.37 | 4.75 | 2.11 | 1.68 | 0.82 | 0.67 | 1.15 |

Note: The amount of element is measured by XRF in the form of oxides.

### 3.2. Screening Sodium Additives

3.2.1. Comparison of Leaching Efficiencies for Different Sodium Additives

In order to further enhance the leaching efficiencies of V and W, different kinds of common sodium additives were investigated. The roasting conditions were as follows: the mass ratio of sodium additive and catalyst was 1:1, the mass ratio of NaCl and NaOH was 1:1, the roasting temperature was 750 °C, and the roasting time remained 2.0 h. The leaching conditions were as follows: the liquid–solid ratio of the roasted clinker and water was 8 mL/g, stirring speed was 200 r/min, leaching temperature was 90 °C, and leaching time was 2.0 h. Table 2 shows the effect of different sodium additives on the efficiencies of vanadium and tungsten. As shown in Table 2, compared with NaOH, Na$_2$CO$_3$, NaCl, and Na$_2$SO$_4$ additives, NaCl and Na$_2$SO$_4$ additives showed a better leaching efficiency of V than the NaOH additive, while they showed a relatively low leaching efficiency of W. Compared with the leaching efficiencies of metals for pure NaOH additive, the leaching efficiencies of V and W for Na$_2$CO$_3$ additive were 90% and 73.25%, which indicated that increasing the alkalinity of sodium additive enhanced to leach W and V. In the three

composite additives (NaOH and Na$_2$CO$_3$, NaOH and NaCl, and NaOH and Na$_2$SO$_4$), the leaching efficiencies of V and W for NaOH and NaCl composite additive were the highest, which can reach 91.39% and 98.26%, respectively.

**Table 2.** The effect of different sodium additives on the leaching efficiencies of vanadium and tungsten.

| Sodium Additive | Leaching Efficiency of V/% | Leaching Efficiency of W/% |
|---|---|---|
| NaOH | 75.04 | 94.43 |
| NaCl | 76.51 | 19.01 |
| Na$_2$SO$_4$ | 76.30 | 17.24 |
| Na$_2$CO$_3$ | 73.25 | 90 |
| NaOH & Na$_2$CO$_3$ | 93.25 | 84.75 |
| NaOH & NaCl | 91.39 | 98.26 |
| NaOH & Na$_2$SO$_4$ | 88.27 | 95.88 |

### 3.2.2. Comparison of Melting Points for Different Sodium Additives

On the other hand, in order to decrease the roasting temperature and further reduce cost, the melting points of several common sodium additives were investigated. Figure 3 shows the melting points of some sodium additives based on some literature. As shown in Figure 3, compared with other sodium additives, NaOH performed the lowest melting point, which indicated that when the roasting temperature was over 400 °C, NaOH presented a molten state so that the roasting reactions shifted to liquid–solid reactions, which can enhance the mass transfer process. Combined with the data in Table 2, we found that the composite additives adding a certain amount of NaOH showed higher leaching efficiencies than these of pure Na$_2$SO$_4$, Na$_2$CO$_3$, or NaCl additives, which can be partly attributed to the promoting effect of NaOH fusant for the roasting reactions. Therefore, NaOH and NaCl can be screened and employed as roasting composite additives.

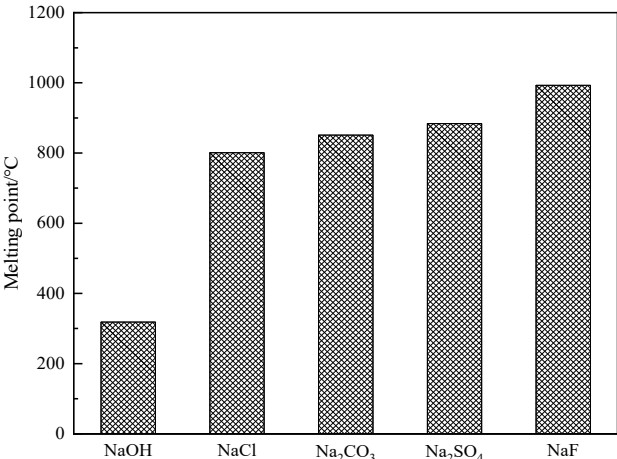

**Figure 3.** The melting points of different sodium additives.

### 3.3. Roasting Parameter Analysis

A single-factor experiment was first adopted to determine the influence of different conditions on the leaching efficiencies of elements, and then, an orthogonal experiment was employed to optimize the roasting conditions. In the following roasting experiment, the leaching conditions should keep consistent. The leaching conditions were as follows: the liquid–solid ratio of roasted clinker and water was 8 mL/g, stirring speed was 200 r/min, leaching temperature was 90 °C, and leaching time was 2.0 h.

### 3.3.1. Roasting Temperature

The roasting temperature is one of the most critical factors affecting the element leaching efficiency [16]. During the roasting step, the mass ratio of sodium additive and catalyst was 1:1, the mass ratio of NaCl and NaOH was 1:1, and the roasting time was 2.0 h. Figure 4 shows that the leaching efficiencies of V and W significantly increased with the increase of the roasting temperature between 450 °C and 750 °C. However, when the roasting temperature exceeded 750 °C, the leaching efficiencies of V and W slightly decreased.

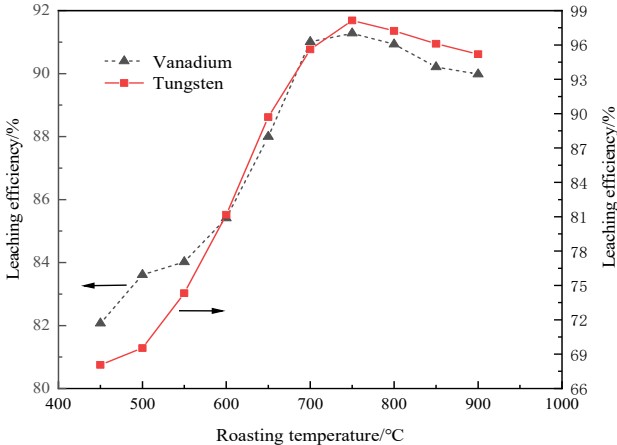

**Figure 4.** The effect of the roasting temperature on the leaching efficiencies of V and W.

At low temperatures, such as 450 °C, owing to that the reactants remained solid and the contact area among different materials was relatively small, resulting in insufficient reaction among different elements, the leaching efficiencies of V and W were only 82.07% and 68.06%, respectively. When the temperature raised to over 500 °C, on the one hand, sodium hydroxide became a molten liquid and roasting reactions converted into liquid-solid reactions, which promoted the contact area among reactants and mass transfer rate among species, thus strengthening the roasting reaction and enhancing the leaching efficiencies of V and W. On the other hand, when some oxides, such as vanadium, tungsten, silicon, and aluminum, were present in the reaction system and the reaction temperature reached over 500 °C, NaCl can decompose into $Na_2O$ and $Cl_2$, for which $Cl_2$ can be employed as catalyst and oxidant to accelerate the oxidation of low valent vanadium, thus vastly enhancing the leaching efficiency of vanadium [35–37]. When the temperature raised to over 800 °C, reactants showed a process of caking, resulting in failing to be leached for partial elements. Considering leaching efficiencies and safety and energy consumption, a roasting temperature between 700 °C and 800 °C can be regarded as the best.

### 3.3.2. Roasting Time

The roasting time is another significant factor influencing the leaching efficiencies of V and W. Figure 5 shows that the effect of roasting time on the leaching efficiencies of V and W under the following conditions: the mass ratio of sodium additive and catalyst was 1:1, the mass ratio of NaCl and NaOH was 1:1, and the roasting temperature was 750 °C. As shown in Figure 5, the leaching efficiencies of V and W increased with the increase of the roasting time between 0.5 h and 2.5 h. When the roasting time was 0.5 h, both the roasting reactions and mass transfer among materials were inadequate, leading to leaching efficiencies of V and W of only 88.72% and 93.45%, respectively. When the roasting time increased from 0.5 h to 2.5 h, owing to that the roasting reactions had been proceeding step by step, the leaching efficiencies of V and W gradually increased. However, when roasting temperature increased to 3.0 h, roasted clinker partly showed caking, which caused leaching of part elements to become more difficult, so the leaching efficiencies of V

and W decreased to 90.07% and 96.05%, respectively. In conclusion, roasting time should remain between 1.5 h and 2.5 h.

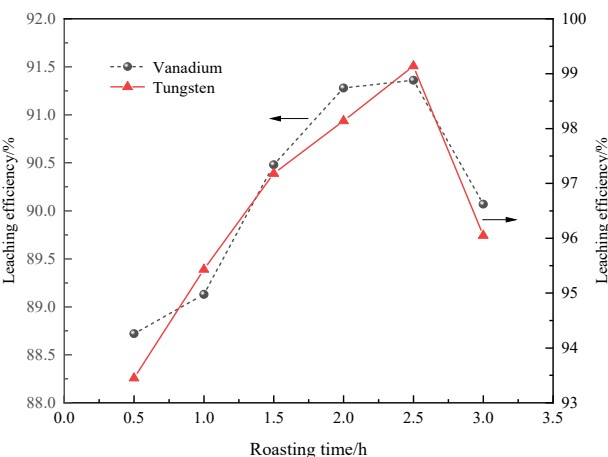

**Figure 5.** The effect of roasting time on the leaching efficiencies of V and W.

### 3.3.3. Mass Ratio of Sodium Additive and Catalyst

Figure 6 shows the effect of the mass ratio of sodium additive and catalyst on the leaching efficiencies of V and W under the following conditions: the mass ratio of NaCl and NaOH was 1:1, the roasting temperature was equal to 750 °C, and the roasting time was 2.0 h. As shown in Figure 6, with the increase of the mass ratio of sodium additive and catalysts, the leaching efficiencies of V and W first increased and then remained unchanged. When the mass ratio of sodium additive and catalyst was relatively low, because reactions between sodium additives and spent catalyst were insufficient, the leaching efficiencies of V and W were only 88.18% and 95.19%, respectively. However, when the mass ratio of sodium additive and catalyst reached over 1.5, the roasting reactions had almost reached balance, increasing the content of sodium additives failed to improve the leaching efficiencies of V and W. Considering leaching efficiencies and cost, a mass ratio of sodium additive and catalysts between 1.5 and 2.5 was found to be optimal.

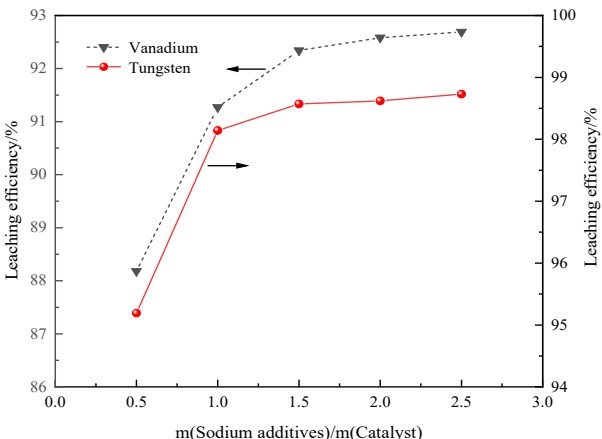

**Figure 6.** The effect of the mass ratio of sodium additive and catalyst on the leaching efficiencies of V and W.

### 3.3.4. Mass Ratio of NaCl and NaOH

Figure 7 shows the effect of the mass ratio of NaCl and NaOH on the leaching efficiencies of V and W when the mass ratio of sodium additive and catalyst was 1:1, the roasting temperature was equal to 750 °C, and the roasting time remained 2.0 h. As shown in Figure 7, with the increase of the mass ratio of NaCl and NaOH, the leaching efficiencies

of V and W first gradually increased, then almost remained the same and finally decreased. When the content of NaCl was zero, the leaching efficiencies of V and W were only 75.04% and 94.43%, respectively. When gradually adding NaCl, one the on hand, $Cl_2$ originating from the part decomposition of NaCl can act as catalyst and oxidant to oxidize low valent vanadium into pentavalent vanadium, and simultaneously promote the roasting reactions [38,39]. On the other hand, compared with NaOH, $Na_2O$ deriving from the part decomposition of NaCl had stronger basicity so that $Na_2O$ can be effective to destroy the catalyst structure and easier to react with V and W, thus increasing the leaching efficiencies of V and W. However, when the mass ratio of NaCl and NaOH increased to over 1.5, owing to the lower content of NaOH with a lower melting point and higher content of NaCl with a higher melting point, molten liquid volume correspondingly decreased at the roasting temperature of 750 °C, resulting in reducing the liquid–solid mass transfer rate and lower leaching efficiencies of V and W. Therefore, the mass ratio of NaCl and NaOH should be between 1.0 and 2.0.

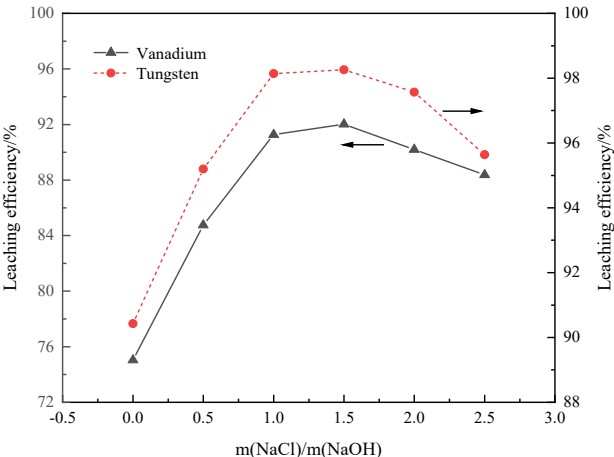

**Figure 7.** The effect of the mass ratio of NaCl and NaOH on the leaching efficiencies of V and W.

### 3.4. Orthogonal Experiment and Weight Matrix Analysis of Roasting Parameter

3.4.1. Orthogonal Experiment

According to the single-factor roasting experiment, three better levels in each of the factors, namely roasting temperature, roasting time, mass ratio of sodium additive and catalyst, and mass ratio of NaCl and NaOH, were screened, respectively. Thereby, these factors were selected as the experimental variables, while the leaching efficiencies of V and W were used as the response values. The four-factor and three-level optimization schemes were designed by Taguchi Design. Nine experimental points were obtained by the Minitab 15 software.

Table 3 shows the orthogonal experiment design and results of leaching vanadium. As shown in Table 3, the range values R of the four factors were 2.15, 0.33, 0.40, and 0.66, respectively. Therefore, the influence of these factors on the leaching efficiency of vanadium was shown as follows, from most significant to least significant: roasting temperature, m(NaCl)/m(NaOH), m(sodium additives)/m(catalyst), and roasting time. Among the index values k1, k2, and k3 corresponding to the three levels of roasting temperature, roasting time, m(sodium additives)/m(catalyst), and m(NaCl)/m(NaOH), k2 of roasting temperature, k3 of roasting time, k3 of m(sodium salts)/m(catalyst), and k2 of m(NaCl)/m(NaOH) were the largest. Therefore, the optimal roasting conditions for leaching vanadium were roasting temperature 750 °C, roasting time 2.5 h, m(sodium additives)/m(catalyst) 2.5, and m(NaCl)/m(NaOH) 1.5.

**Table 3.** Orthogonal experiment design and results of leaching vanadium.

| Standard Order | Roasting Temperature/°C (A) | Roasting Time/h (B) | m(Sodium Additives) /m(Catalyst) (C) | m(NaCl) /m(NaOH) (D) | Leaching Efficiency of Vanadium/% |
|---|---|---|---|---|---|
| 1 | 700 | 1.5 | 1.5 | 1.0 | 89.78 |
| 2 | 700 | 2.0 | 2.0 | 1.5 | 91.06 |
| 3 | 700 | 2.5 | 2.5 | 2.0 | 91.16 |
| 4 | 750 | 1.5 | 2.0 | 2.0 | 92.88 |
| 5 | 750 | 2.0 | 2.5 | 1.0 | 92.65 |
| 6 | 750 | 2.5 | 1.5 | 1.5 | 92.92 |
| 7 | 800 | 1.5 | 2.5 | 1.5 | 92.32 |
| 8 | 800 | 2.0 | 1.5 | 2.0 | 92.24 |
| 9 | 800 | 2.5 | 2.0 | 1.0 | 91.88 |
| k1 | 90.67 | 91.66 | 91.65 | 91.44 | |
| k2 | 92.82 | 91.98 | 91.94 | 92.10 | |
| k3 | 92.15 | 91.99 | 92.04 | 92.09 | |
| R | 2.15 | 0.33 | 0.40 | 0.66 | |

Table 4 shows the orthogonal experiment design and results of leaching tungsten. As shown in Table 4, the range values R of four factors were 2.87, 0.62, 0.22, and 0.92, respectively. Therefore, the influence of these factors on the leaching efficiency of tungsten was as follows, from most significant to least significant: roasting temperature, m(NaCl)/m(NaOH), roasting time, and m(sodium additives)/m(catalyst). Among the index values k1, k2, and k3 corresponding to the three levels of roasting temperature, roasting time, m(sodium additives)/m(catalyst), and m(NaCl)/m(NaOH), k2 of roasting temperature, k3 of roasting time, k2 of m(sodium additives)/m(catalyst), and k2 of m(NaCl)/m(NaOH) were the largest. Therefore, the optimal roasting conditions for leaching tungsten were roasting temperature 750 °C, roasting time 2.5 h, m(sodium additives)/m(catalyst) 2.0, and m(NaCl)/m(NaOH) 1.5.

**Table 4.** Orthogonal experiment design and results of leaching tungsten.

| Standard Order | Roasting Temperature/°C (A) | Roasting Time/h (B) | m(Sodium Additives) /m(Catalyst) (C) | m(NaCl) /m(NaOH) (D) | Leaching Efficiency of Tungsten/% |
|---|---|---|---|---|---|
| 1 | 700 | 1.5 | 1.5 | 1.0 | 93.81 |
| 2 | 700 | 2.0 | 2.0 | 1.5 | 95.18 |
| 3 | 700 | 2.5 | 2.5 | 2.0 | 95.22 |
| 4 | 750 | 1.5 | 2.0 | 2.0 | 97.59 |
| 5 | 750 | 2.0 | 2.5 | 1.0 | 97.01 |
| 6 | 750 | 2.5 | 1.5 | 1.5 | 98.21 |
| 7 | 800 | 1.5 | 2.5 | 1.5 | 97.25 |
| 8 | 800 | 2.0 | 1.5 | 2.0 | 97.15 |
| 9 | 800 | 2.5 | 2.0 | 1.0 | 97.07 |
| k1 | 94.74 | 96.22 | 96.39 | 95.96 | |
| k2 | 97.60 | 96.45 | 96.61 | 96.88 | |
| k3 | 97.16 | 96.83 | 96.50 | 96.66 | |
| R | 2.87 | 0.62 | 0.22 | 0.92 | |

3.4.2. Weight Matrix Analysis

According to the above orthogonal test results, it was not difficult to find that the optimal experiment conditions of leaching vanadium was different from that of leaching tungsten. Owing to the fact that the above orthogonal experiment had two indexes, in order to realize the comprehensive evaluation of various factors and further optimize reaction conditions, it is necessary to investigate the weight of each index affecting the experimental

results. As a result, the weight matrix analysis was introduced into the multiple indexes orthogonal experiment design.

In this the weight matrix analysis, a three-tier data model needed to be established based on the orthogonal experiment results [40]. Table 5 was the a three-tier data model. As shown in Table 5, the first layer is the test index layer, the second layer was the factor layer, and the third layer was the level layer [41].

**Table 5.** The data structure of the orthogonal experiment.

| Test Index Layer | Test Index | | | |
|---|---|---|---|---|
| Factor layer | Factor $H_1$ | Factor $H_2$ | $\ldots$ | Factor $H_h$ |
| Level layer | $H_{11} H_{12} H_{13} H_{1m}$ | $H_{21} H_{22} H_{2m}$ | $\ldots$ | $H_{h1} H_{h2} H_{hm}$ |

According to the experimental data, the definition of the matrix was given, as shown below.

1. test index layer matrix

If the orthogonal experiment had m factors and h levels, $k_{ij}$ was the average value of test indexes in the j level of factor $A_i$, and N was the test index layer matrix, establishing matrix (10):

$$N = \begin{bmatrix} k_{11} & 0 & 0 & \cdots & 0 \\ k_{12} & 0 & 0 & \cdots & 0 \\ \cdots & \cdots & \cdots & \cdots & \cdots \\ k_{1m} & 0 & 0 & \cdots & 0 \\ 0 & k_{21} & 0 & \cdots & 0 \\ 0 & k_{22} & 0 & \cdots & 0 \\ \cdots & \cdots & \cdots & \cdots & \cdots \\ 0 & k_{2m} & 0 & \cdots & 0 \\ \cdots & \cdots & \cdots & \cdots & \cdots \\ 0 & 0 & 0 & \cdots & k_{h1} \\ 0 & 0 & 0 & \cdots & k_{h3} \\ \cdots & \cdots & \cdots & \cdots & \cdots \\ 0 & 0 & 0 & \cdots & k_{hm} \end{bmatrix} \tag{10}$$

2. factor layer matrix

Let $T_i = 1 / \sum\limits_{j=1}^{m} k_{ij}$, where T is the factor layer matrix, establishing matrix (11):

$$T = \begin{bmatrix} T_1 & 0 & 0 & 0 \\ 0 & T_2 & \cdots & 0 \\ \cdots & \cdots & \cdots & \cdots \\ 0 & 0 & 0 & T_h \end{bmatrix} \tag{11}$$

3. level layer matrix

The range value in the orthogonal test was R. Let $P_i = R_i / \sum\limits_{i=1}^{h} R_i$, where P is the level layer matrix, establishing matrix (12):

$$P = \begin{bmatrix} P_1 \\ P_2 \\ \cdots \\ P_h \end{bmatrix} \tag{12}$$

4. weight matrix of test index matrix

The weight matrix of test index was defined as Z = NTP, shown as matrix (13):

$$Z = NTP = \begin{bmatrix} A_1 \\ A_2 \\ A_3 \\ B_1 \\ B_2 \\ B_3 \\ C_1 \\ C_2 \\ C_3 \\ D_1 \\ D_2 \\ D_3 \end{bmatrix} \tag{13}$$

where Z is the weight matrix of test index matrix, N is the test index layer matrix, P is the level layer matrix, and A, B, C, D are the factors.

5. Calculation of weight matrix for the orthogonal test

Based on the results of the orthogonal experiment and the definition of the weight matrix, the weight matrix analysis of leaching vanadium or tungsten was conducted. The analysis process is as follows:

(1) weight matrix calculation of test indexes for leaching vanadium

$$N1 = \begin{bmatrix} 90.67 & 0 & 0 & 0 \\ 92.82 & 0 & 0 & 0 \\ 92.15 & 0 & 0 & 0 \\ 0 & 91.66 & 0 & 0 \\ 0 & 91.98 & 0 & 0 \\ 0 & 91.99 & 0 & 0 \\ 0 & 0 & 91.65 & 0 \\ 0 & 0 & 91.94 & 0 \\ 0 & 0 & 92.04 & 0 \\ 0 & 0 & 0 & 91.44 \\ 0 & 0 & 0 & 92.10 \\ 0 & 0 & 0 & 92.09 \end{bmatrix}$$

$$T1 = \begin{bmatrix} \frac{1}{275.63} & 0 & 0 & 0 \\ 0 & \frac{1}{275.63} & 0 & 0 \\ 0 & 0 & \frac{1}{275.63} & 0 \\ 0 & 0 & 0 & \frac{1}{275.63} \end{bmatrix}$$

$$P1 = \begin{bmatrix} \frac{2.15}{3.54} \\ \frac{0.34}{3.54} \\ \frac{0.39}{3.54} \\ \frac{0.66}{3.54} \end{bmatrix}$$

$$
Z_1 = \begin{bmatrix}
90.67 & 0 & 0 & 0 \\
92.82 & 0 & 0 & 0 \\
92.15 & 0 & 0 & 0 \\
0 & 91.66 & 0 & 0 \\
0 & 91.98 & 0 & 0 \\
0 & 91.99 & 0 & 0 \\
0 & 0 & 91.65 & 0 \\
0 & 0 & 91.94 & 0 \\
0 & 0 & 92.04 & 0 \\
0 & 0 & 0 & 91.44 \\
0 & 0 & 0 & 92.10 \\
0 & 0 & 0 & 92.09
\end{bmatrix}
\begin{bmatrix}
\frac{1}{275.63} & 0 & 0 & 0 \\
0 & \frac{1}{275.63} & 0 & 0 \\
0 & 0 & \frac{1}{275.63} & 0 \\
0 & 0 & 0 & \frac{1}{275.63}
\end{bmatrix}
\begin{bmatrix}
\frac{2.15}{3.54} \\
\frac{0.34}{3.54} \\
\frac{0.39}{3.54} \\
\frac{0.66}{3.54}
\end{bmatrix}
$$

$$
= \begin{bmatrix}
0.199816 \\
0.204554 \\
0.203035 \\
0.031935 \\
0.032051 \\
0.032054 \\
0.036631 \\
0.036753 \\
0.036786 \\
0.061842 \\
0.062298 \\
0.062291
\end{bmatrix}
$$

(2) weight matrix calculation of test indexes for leaching tungsten

$$
N_2 = \begin{bmatrix}
94.74 & 0 & 0 & 0 \\
97.60 & 0 & 0 & 0 \\
97.16 & 0 & 0 & 0 \\
0 & 96.22 & 0 & 0 \\
0 & 96.45 & 0 & 0 \\
0 & 96.83 & 0 & 0 \\
0 & 0 & 96.39 & 0 \\
0 & 0 & 96.61 & 0 \\
0 & 0 & 96.49 & 0 \\
0 & 0 & 0 & 95.96 \\
0 & 0 & 0 & 96.88 \\
0 & 0 & 0 & 96.65
\end{bmatrix}
\quad
T2 = \begin{bmatrix}
\frac{1}{289.50} & 0 & 0 & 0 \\
0 & \frac{1}{289.50} & 0 & 0 \\
0 & 0 & \frac{1}{289.50} & 0 \\
0 & 0 & 0 & \frac{1}{289.50}
\end{bmatrix}
$$

$$
P_2 = \begin{bmatrix}
\frac{2.87}{4.63} \\
\frac{0.62}{4.63} \\
\frac{0.22}{4.63} \\
\frac{0.92}{4.63}
\end{bmatrix}
$$

$$
Z2 = \begin{bmatrix}
94.74 & 0 & 0 & 0 \\
97.60 & 0 & 0 & 0 \\
97.16 & 0 & 0 & 0 \\
0 & 96.22 & 0 & 0 \\
0 & 96.45 & 0 & 0 \\
0 & 96.83 & 0 & 0 \\
0 & 0 & 96.39 & 0 \\
0 & 0 & 96.61 & 0 \\
0 & 0 & 96.49 & 0 \\
0 & 0 & 0 & 95.96 \\
0 & 0 & 0 & 96.88 \\
0 & 0 & 0 & 96.65
\end{bmatrix}
\begin{bmatrix}
\frac{1}{289.50} & 0 & 0 & 0 \\
0 & \frac{1}{289.50} & 0 & 0 \\
0 & 0 & \frac{1}{289.50} & 0 \\
0 & 0 & 0 & \frac{1}{289.50}
\end{bmatrix}
\begin{bmatrix}
\frac{2.87}{4.63} \\
\frac{0.62}{4.63} \\
\frac{0.22}{4.63} \\
\frac{0.92}{4.63}
\end{bmatrix}
$$

$$
= \begin{bmatrix}
0.202884 \\
0.208958 \\
0.208029 \\
0.044511 \\
0.044619 \\
0.044793 \\
0.015818 \\
0.015856 \\
0.015832 \\
0.065870 \\
0.066486 \\
0.066347
\end{bmatrix}
$$

(3) total weight matrix of orthogonal test

The total weight matrix of the orthogonal test was the average value of two test index matrices, as follows:

$$
Z = \frac{Z_1 + Z_2}{2} = \frac{1}{2}\left\{
\begin{bmatrix}
0.199816 \\
0.204554 \\
0.203035 \\
0.031935 \\
0.032051 \\
0.032054 \\
0.036631 \\
0.036753 \\
0.036786 \\
0.061842 \\
0.062298 \\
0.062291
\end{bmatrix}
+
\begin{bmatrix}
0.202884 \\
0.208958 \\
0.208029 \\
0.044511 \\
0.044619 \\
0.044793 \\
0.015818 \\
0.015856 \\
0.015832 \\
0.065870 \\
0.066486 \\
0.066347
\end{bmatrix}
\right\}
=
\begin{bmatrix}
0.201350 \\
0.206756 \\
0.205532 \\
0.038223 \\
0.038335 \\
0.038424 \\
0.026225 \\
0.026305 \\
0.026309 \\
0.063856 \\
0.064392 \\
0.064319
\end{bmatrix}
=
\begin{bmatrix}
A_1 \\
A_2 \\
A_3 \\
B_1 \\
B_2 \\
B_3 \\
C_1 \\
C_2 \\
C_3 \\
D_1 \\
D_2 \\
D_3
\end{bmatrix}
$$

It can be seen from the above calculation that $A_2$ was the largest among the total weight values of factor A, $B_3$ was the largest among the total weight values of factor B, $C_3$ was the largest among the total weight values of factor C, and $D_2$ was the largest among the total weight values of factor D. Therefore, the optimal roasting conditions are as follows: the roasting temperature was 750 °C, the roasting time was 2.5 h, the mass ratio of sodium salts and catalyst was 2.5, and the mass ratio of NaCl and NaOH was 1.5.

*3.5. Repeated Test*

In order to acquire the highest possible leaching efficiencies of metals, we investigated the leaching efficiencies of V and W under the above optimal roasting conditions. Figure 8 showed the leaching efficiencies of V and W under the optimal roasting conditions. As shown in Figure 8, it can be seen that the leaching efficiencies of V and W almost kept unchanged in the results of seven repeated tests. Under the optimal roasting conditions, the leaching efficiencies of V and W can reach 93.25% and 99.17%, respectively.

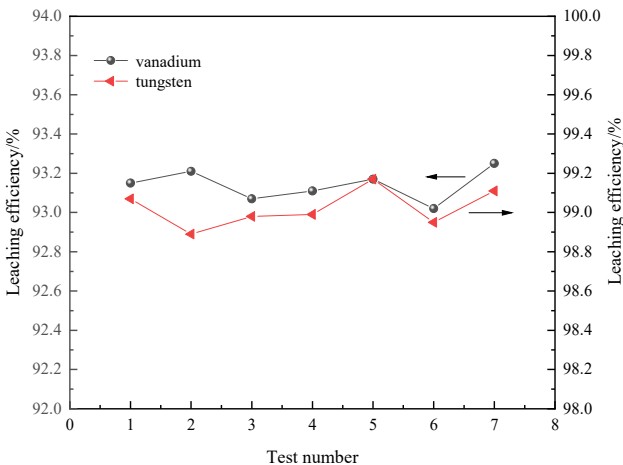

**Figure 8.** The leaching efficiencies of V and W under the optimal roasting conditions.

*3.6. XRD Analysis of Roasted Clinker*

Figure 9 shows the XRD spectra of roasted clinker at different roasting temperatures. According to the XRF results, the spent SCR catalyst simultaneously contained vanadium element and sulfur element. Some literature showed that $V_2O_5$ was partly reduced into low-valent vanadate salts ($VOSO_4$) during the SCR catalyst's operation, namely that the active component vanadium in the spent SCR catalyst mainly existed in the form of $V_2O_5$ and $VOSO_4$ [26,32]. As shown in Figure 9, at a low temperature (450 °C), the roasted clinker contained a lot of vanadium dioxide ($VO_2$) deriving from the composition of $VOSO_4$ in the reaction system, while it failed to show the diffraction peak of sodium metavanadate ($NaVO_3$) in the XRD spectra, which illustrated that the amount of generative $NaVO_3$ was lower and tetravalent vanadium was difficult to occur the roasting reactions. Moreover, a little amount of vanadium existed in the form of titanium-vanadium oxide (($Ti_{0.5}V_{0.5})_2O_3$), i.e., a small amount of vanadium embedded in the framework of titanium oxide, the leaching of which failed. With the increase of roasting temperature, the intensity of the $VO_2$ diffraction peak gradually became weaker; at the same time, a series of $NaVO_3$ diffraction peaks began to appear, which inferred that the $VO_2$ may first oxidize into $V_2O_5$ and then react with sodium additives. At 450 °C, titanium element mainly existed in the form of $TiO_2$ and only a few titanium elements existed in the form of $Na_2Ti_3O_7$, while with the increase in temperature, more $TiO_2$ began to anticipate in the roasting reaction and converted into $Na_2Ti_3O_7$, which showed that a part of the Ti-O octahedrons structure in the catalyst was destroyed. In addition, only a few $Na_2WO_4$ diffraction peaks occurred at 450 °C and the tungsten element failed to form composite oxide at all the temperature ranges, which implied that the tungsten element was only wrapped in the spent SCR catalyst structure. With the increase of temperature, owing to that the Ti-O octahedrons structure was destroyed, more tungsten elements were released from Ti-O octahedrons structure and started to undergo the roasting reactions, and thus, more $Na_2WO_4$ diffraction peaks appeared in the XRD spectra.

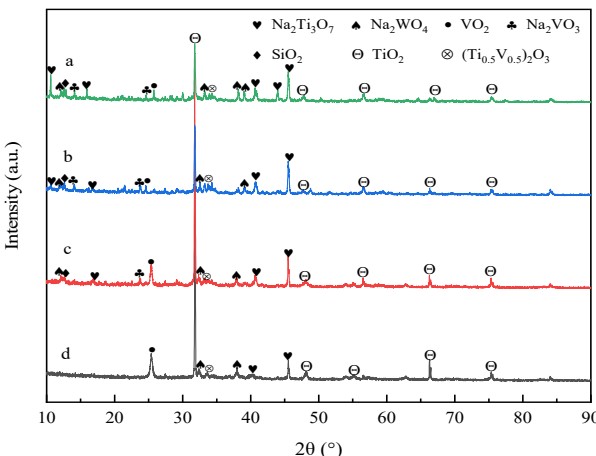

**Figure 9.** The XRD spectra of roasted clinker at different roasting temperatures: (a) 750 °C, (b) 650 °C, (c) 550 °C, and (d) 450 °C.

### 3.7. Comparison of Vanadium and Tungsten Leaching Efficiencies for Different Processes

Based on the optimization of the roasting conditions in the orthogonal experiment, we determined the highest leaching efficiencies of V and W by a repeated test, after which the leaching efficiencies of metals in the different prcocesses were compared. Table 6 shows the leaching efficiencies of V and W for different processes. As shown in Table 6, compared with alkali or acid methods, the roasting method had a higher and stable leaching efficiency of W, while the process of pure $Na_2CO_3$ roasting and water leaching showed lower leaching efficiency of V. In the alkali methods, the leaching efficiencies of V and W for NaOH direct leaching were close to 90%, and although it was reported that the leaching efficiencies of V and W for $(NH_4)_2CO_3$ leaching reached 98% and 99%, $(NH_4)_2CO_3$ showed poor stability when the solution was heated so that the liquid–solid ratio needed to be relatively high, which meant that large amounts of waste water can be produced. The acid methods were generally employed to extract V from spent SCR catalyst, for the purpose of simplifying the steps of recycling vanadium, and the leaching efficiency of V was approximately 85%. In our approach, the use of NaOH-NaCl composite additives was benefical to the leaching process. Metals can be effectively leached, and the leaching efficiencies of V and W can reach 93.25% and 99.17, respectively. Simultaneously, it can also be seen from Table 6 that adding NaOH with a lower melting point into the higher melting point additives can decrease the roasting temperature.

**Table 6.** The leaching efficiencies of V and W for different processes.

| No. | Process | Roasting Conditions | Leaching Conditions | Leaching Efficiency of V/% | Leaching Efficiency of W/% | Ref. |
|-----|---------|---------------------|---------------------|----------------------------|----------------------------|------|
| 1 | $Na_2CO_3$ roasting and water leaching | 900 °C, 2.0 h, and 30 wt.% $Na_2CO_3$ | 3:1 L/S mass ratio, 40 °C and 3.0 h | 49.05 | 99.02 | [32] |
| 2 | $Na_2CO_3$-NaCl roasting and water leaching | 750 °C, 2.0 h, 0.5:1.1 $NaCl/Na_2CO_3$ mole ratio and 1.2 $Na_2CO_3$/ catalyst mole ratio | 8:1 L/S mass ratio, 40 °C and 1.0 h | none | 99.10 | [42] |

**Table 6.** *Cont.*

| No. | Process | Roasting Conditions | Leaching Conditions | Leaching Efficiency of V/% | Leaching Efficiency of W/% | Ref. |
|---|---|---|---|---|---|---|
| 3 | NaOH-NaCl roasting and water leaching | 750 °C, 2.5 h, 1.5 NaCl/Na$_2$CO$_3$ mass ratio and 2.5 additives/catalyst mass ratio | 6:1 L/S mass ratio, 90 °C and 1.0 h | 93.25 | 99.17 | this work |
| 4 | NaOH directly leaching | none | 3 M, 2.5 mL/g L/S ratio, 250 °C, and 2.0 h | 92 | 87 | [14] |
| 5 | NaOH directly leaching | none | 3% pulp density 0.3 L/S mass ratio, 90 °C, and 30 min | 87 | 91 | [22] |
| 6 | (NH$_4$)$_2$CO$_3$ leaching | none | 3.0 M (NH$_4$)$_2$CO$_3$, 1.5 M H$_2$O$_2$, 25:1 L/S ratio 70 °C, and 30 min | 98 | 99 | [15] |
| 7 | H$_2$SO$_4$ leaching | none | 45 wt.% with Na$_2$SO$_3$, 12 mL/g L/S ratio, 100 °C, and 180 min | 85 | none | [26] |
| 8 | H$_2$C$_2$O$_2$ leaching | none | 20 mL/g L/S ratio 1.0 M, 90 °C, and 180 min | 84.22 | none | [7] |

## 4. Conclusions

In this study, a process of composite roasting and water leaching to extract vanadium and tungsten from spent SCR catalyst was screened. Among different kinds of sodium additives, NaOH-NaCl composite roasting performed higher leaching efficiencies of V and W. On the one hand, adding NaOH, which has a lower melting point, can decrease the roasting temperature by enhancing mass transfer among different matters. On the other hand, Cl$_2$ originating from the decomposition of NaCl can be used as catalyst and oxidant to promote the oxidation of low valent vanadium, which enhanced the leaching of vanadium.

In the single-factor experiment, it was found that with the increase of roasting temperature, roasting time, and mass ratio of NaCl and NaOH, the leaching efficiencies of metals first increased and then decreased, while with the increase of mass ratio of sodium additive and catalyst, the leaching efficiencies of metals first increased and were then almost unchanged. Different roasting conditions were investigated and optimized by the orthogonal experiment and weight matrix analysis. The results showed that the roasting temperature was the most important factor influencing the leaching efficiencies of V and W. The best roasting conditions were as follows: the roasting temperature was 750 °C, the roasting time was 2.5 h, the mass ratio of sodium salts and catalyst was 2.5, and the mass ratio of NaCl and NaOH was 1.5. The optimal leaching efficiencies of V and W were 93.25% and 99.17%, respectively.

The XRD analysis showed that with the increase of roasting temperature, the structure of the catalyst was destroyed, releasing more vanadium and tungsten from inside the catalyst and promoting the roasting reactions. Lastly, we compared the leaching efficiencies of metals for several processes of recycling spent catalyst, such as roasting methods, acid methods, and alkali methods. The NaOH-NaCl composite roasting and water leaching kept the highest leaching efficiencies of V and W, which were 93.25% and 99.17, while the leaching efficiencies of V and W for Na$_2$CO$_3$ roasting and water leaching were 49.05% and 99.02%, the leaching efficiencies of V and W for alkali leaching were close 90% and the leaching efficiency of V for acid leaching was close to 85%. Compared with the traditional process of pure soda roasting and water leaching, while maintaining better or improved efficiencies of metals, the roasting temperature was cut down by 150 °C.

**Author Contributions:** Experimental preparation and operation, writing—original draft, and employing software, B.W.; writing—review and editing, Q.Y. All authors have read and agreed to the published version of the manuscript.

**Funding:** This work has no funding support.

**Institutional Review Board Statement:** Not applicable.

**Informed Consent Statement:** Not applicable.

**Data Availability Statement:** Not applicable.

**Conflicts of Interest:** The authors declare no conflict of interest.

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
