# Peer review of "Optimization of Roasting Parameters for Recovery of Vanadium and Tungsten from Spent SCR Catalyst with Composite Roasting"

_processes, doi:10.3390/pr9111923_

Round 1
Reviewer 1 Report
The paper is clear and well written.
The topic of the paper is appropriate for the Journal.
The text of the paper was written correctly in terms of stylistically, punctuation and terminology.
The paper was correctly edited and graphically developed at a good level.
The literature was chosen correctly and fully used in the paper.
I do not see any shortcuts.
The paper deserves a positive assessment because it is current and interesting from both a cognitive and practical point of view.
Author Response
Dear reviewer,
Thank you for taking the time to review my manuscript processes-1427548-org. Based on your review comments, I have made some modifications as following.
- In the part of abstract, I revised the purpose and the main results that achieved with my study in detail.
- In the part of introduction, I added some background explanation about spent SCR catalyst recovery in detail and perfected objectives of the study need to be addressed.
- In the part of materials and methods, I added two pictures about the macroscopic morphology of spent SCR catalyst. I added greater detail the experimental processes in the research.
- In the part of results and discussion, (1) I added a XRD diffraction pattern of spent SCR catalyst in 3.1 to explain the minerals phase of spent catalyst. From the XRD diffraction pattern of spent SCR catalyst, I found that the main structure of spent SCR catalyst was anatase TiO2, and owing to the less content and the uniform distribution of V and W, the diffraction peaks of V and W failed to appear in the figure 1. (2) I added a section to compare the melting points of different sodium additives in 3.2.2. From the melting points of different sodium additives in figure 2, it can be seen that NaOH had the lowest melting point, which indicated that adding NaOH into the higher melting point additives can enhance the mass transfer among different materials, thus decreasing the roasting temperature. (3) I added a discussion section to compare with the leaching efficiencies of V and W for other processes obtained by other studies and other authors in 3.7. From the leaching efficiencies of V and W for different processes in table 6, the efficiency of V for the NaOH-NaCl composite roasting was much higher than that for pure soda roasting, and simultaneously the efficiency of W for the NaOH-NaCl composite roasting was the same as that for other sodium roasting methods. Moreover, the roasting temperature of NaOH-NaCl composite roasting cut down by 150 ℃ than that of pure soda roasting. In addition, the efficiency of V and W for the NaOH-NaCl composite roasting (93.25% leaching efficiency of V and 99.17% leaching efficiency of W) were also higher than that for the acid methods (about 85% leaching efficiency of V) or alkali methods (about 92% leaching efficiency of V and 87% leaching efficiency of W). (4) At last, I revised the format and language and inserted a table 5.
- In the part of conclusion, I explained the main results achieved in this research in detail.
Finally, there are still many deficiencies in my manuscript. Owing to being limited by the modification time and scientific research conditions of my university, I failed to finish the XPS characterization, BET characterization and SEM characterization, and I have only corrected most of the deficiencies in this paper. I hope you understand. Thank you for your review!
Bo Wang
2021.10.20

Reviewer 2 Report
Overall, it's been well written, logical, and reader-friendly, but it would get even better if a few points as shown below are properly addressed.
1. Abstract
The abstract can be confusing to a potential reader if they read only abstract, as ‘vanadium’ and ‘tungsten’ don’t appear in the abstract but only V and W. Adding the words before the use of the chemical symbols would be appropriate helping better understanding of potential readers.
2. There is some information that is missing, such as
2 (a). Previous studies on the SCR catalyst using sodium additives.
2 (b). Underlying physics of the process suggested here
(for example, between 2.1. Materials and Characteristics and 2.2. Experimental Procedure, there should be another chapter addressing requirement for a certain chemical material for what reason, and what reaction is required and expected to occur in what condition, before the experimental procedure is described, or even briefly in the early part of 2.2.)
2 (c). and then, in the result as well. It mainly describes a result of experimental design and a certain condition with a certain material that shows the best outcome, but there should be some more explanations from a physico-chemical point of view, addressing a possible reason for the different results between the different sodium additives, temperatures, etc analyzing the results from that perspective.
3. Table 1 says Chemical composition of ‘fresh’ and ‘spent’ SCR catalyst, but which part of the table is for the fresh SCR and which is for the other?
Author Response
Dear reviewer,
Thank you for taking the time to review my manuscript processes-1427548-org. According to your review comments, I have made some modifications as following.
- In the part of abstract, I revised the format, the purpose and the main results that achieved with my study in detail.
- In the part of introduction, I added some background explanation about spent SCR catalyst recovery in detail and perfected objectives of the study need to be addressed.
- In the part of materials and methods, I added two pictures about the macroscopic morphology of spent SCR catalyst and explained the spent SCR catalyst used in our study. I added greater detail the experimental processes in the research. At the same time, I added some possible roasting reactions in the part of results and discussion 3.1.
- In the part of results and discussion, (1) I added a discussion section to compare with the leaching efficiencies of V and W for other processes obtained by other studies and other authors in 3.7. From the leaching efficiencies of V and W for different processes in table 6, the efficiency of V for the NaOH-NaCl composite roasting was much higher than that for pure soda roasting, and simultaneously the efficiency of W for the NaOH-NaCl composite roasting was the same as that for other sodium roasting methods. Moreover, the roasting temperature of NaOH-NaCl composite roasting cut down by 150 ℃ than that of pure soda roasting. In addition, the efficiency of V and W for the NaOH-NaCl composite roasting (93.25% leaching efficiency of V and 99.17% leaching efficiency of W) were also higher than that for the acid methods (about 85% leaching efficiency of V) or alkali methods (about 92% leaching efficiency of V and 87% leaching efficiency of W). (2) I added a XRD diffraction pattern of spent SCR catalyst in 3.1 to explain the minerals phase of spent catalyst. From the XRD diffraction pattern of spent SCR catalyst, I found that the main structure of spent SCR catalyst was anatase TiO2, and owing to the less content and the uniform distribution of V and W, the diffraction peaks of V and W failed to appear in the figure 1. (3) I added a section to compare the melting points of different sodium additives in 3.2.2. From the melting points of different sodium additives in figure 2, it can be seen that NaOH had the lowest melting point, which indicated that adding NaOH into the higher melting point additives can enhance the mass transfer among different materials, thus decreasing the roasting temperature. (4) At last, I revised the format, language and inserted a table 5.
- In the part of conclusion, I explained the main results achieved in this research in detail.
- The catalysts in table 1 were the spent SCR catalysts. "Fresh and spent SCR catalysts " referred to the catalysts being discarded in a coal-fired power plant just now. It was easy to produce ambiguity here. Therefore, I changed it to "spent SCR catalyst".

Reviewer 3 Report
Reconsider after major revision.
Thank you by invitation to review manuscript number “processes-1427548 - Optimization of
roasting parameters for recovery of vanadium and
tungsten from spent SCR catalyst with composite roasting - Bo Wang and Qiaowen Yang ” for
publication in Journal processes.
The experimental results showed that roasting temperature had the most significant effect on the
leaching efficiencies of V and W, and the optimal roasting conditions were shown as following: the
roasting temperature was 750 , the roasting time was 2.5 h, the mass ℃ ratio of sodium additives and
catalyst was 2.5, and the mass ratio of NaCl and NaOH was 1.5. The results of XRD analysis
showed that increasing temperature promoted roasting reactions, thus enhancing the leaching of V
and W. The orthogonal experiment and weight matrix analysis in our study provided a reference
with the optimization of reaction conditions for multiple indexes experiment.
It may look interesting for publication, however, I suggest that the article be submitted for a new
reviewer.
REVIEWER REPORT(S)
→ Please, improvement the introduction;
→ Please, write a subsection about catalytic activity;
→ Why matrix ?
Please, show graphical representations:
1. Concentration/ppm x Temperauture (K)
2. Absorvance /a.u. x wavenumber (cm-1)
3. Show a table whith thermal stabilities;
4. Show patterns of the catalysts (Intensity (a.u.) x degree);
5. Show TEM images, HRTM images;
6. Volume absorved x relative pressure;
7. Intensity x Binding energy (eV);
8. Intensity x Temperature
9. Stability and tolerance: conversion x Time (min)
Author Response
Dear reviewer,
Thank you for taking the time to review my manuscript processes-1427548-org. Based on your review comments, I have made some modifications as following.
- In the part of abstract, I revised the purpose and the main results that achieved with my study in detail.
- In the part of introduction, I added some background explanation about spent SCR catalyst recovery in detail and perfected objectives of the study need to be addressed.
- My article was about the optimization of roasting conditions for recovery of spent SCR catalyst. I added some descriptions of spent SCR catalyst in materials and methods 2.1. Generally, the spent SCR catalysts meant that the NOx conversion efficiency of the SCR catalyst was less than 50%. We obtained spent SCR catalysts from a coal-fired power plant in Shanghai, China. The data about specific catalyst activity was not given in a coal-fired power plant in Shanghai. Moreover, the catalyst activity had little to do with the content of my paper. Therefore, I thought catalytic activity wasn’t necessary to list in my article.
- In order to effectively solve the problem of scheme optimization with multi-index orthogonal test design, the influence degree of each experimental factor on each index was calculated by matrix method, and then according to the weight of different indexes, the best experimental scheme was determined. In our study, because orthogonal test had two response values (leaching efficiencies of V and W), the weight matrix method was utilized to acquire the optimal reaction conditions by comparing the weights of each response values. The leaching efficiency of V were as important as the leaching efficiency of W so that the total matrix was equal to the mean value of the weight matrix of the orthogonal test results for leaching V and W. Based on the results of total matrix, the optimal test scheme was obtained.
- (1) The relationship between leaching solution concentration and leaching temperature wasn’t discussed in my article. The chart didn’t need to be listed.
(2) Because we used an ultraviolet spectrophotometer to determine the concentration of vanadium or tungsten at 420 nm or 430 nm in the leaching solution, the relationship between absorbance and wavelength wasn’t be obtained.
(3) The thermal stability of catalysts had little to do with the content of my paper. I thought it didn’t need to be listed.
(4) I added a XRD diffraction pattern of spent SCR catalyst in 3.1 to explain the minerals phase of spent catalyst. From the XRD diffraction pattern of spent SCR catalyst, I found that the main structure of spent SCR catalyst was anatase TiO2, and owing to the less content and the uniform distribution of V and W, the diffraction peaks of V and W failed to appear in the figure 1.
(5) I’m sorry that owing to being limited by the modification time and scientific research conditions of my university, I failed to finish the TEM characterization. I added two I added two pictures about the macroscopic morphology of spent SCR catalyst.
(6) I’m sorry that owing to being limited by the modification time and scientific research conditions of my university, I failed to finish the BET characterization.
(7) I’m sorry that owing to being limited by the modification time and scientific research conditions of my university, I failed to finish the XPS characterization.
(8) In my manuscript, I showed the XRD characterization of the relationship between the roasted clinker and roasting temperature in results and discussion 3.7.
(9) I added some descriptions of spent SCR catalyst in materials and methods 2.1. Generally, the spent SCR catalysts meant that the NOx conversion efficiency of the SCR catalyst was less than 50%. The data about specific catalyst activity was not given in a coal-fired power plant in Shanghai. Moreover, the catalyst activity had little to do with the content of my paper. Therefore, I thought catalytic activity wasn’t necessary to list in my article.
Finally, there are still many deficiencies in my manuscript. Owing to being limited by the modification time and scientific research conditions of my university, I failed to finish the XPS characterization, BET characterization and SEM characterization, and I have only corrected most of the deficiencies in this paper. I hope you understand. Thank you for your review!
Reviewer 4 Report
The article presents a topic about the Optimization of roasting parameters for recovery of vanadium and tungsten from spent SCR catalyst with composite roasting, however there are points that need to be clarified and summarized.
- Abstract: I suggest improving the abstract, making clear the purpose and the main results that have been achieved with this study.
- Introduction: The introduction is very general and very short. It should be improved and clearly explain the purpose and final objective of this study with appropriate references. In a research paper, it is expected that introduction section briefly explains the starting background and, even more important, the originality (novelty) and relevancy of the study is well established. Once this is done, hypothesis and objectives of the study need to be addressed, as well as a brief justification of the conducted methodology.
- Experimental procedure: Explain in greater detail the experimental processes carried out in the research.
- Discussion Section: Create a separate Discussion section. The Discussion section should compare the study by clearly comparing the results obtained by the authors with other studies conducted by other authors.
- Conclusions Section: Improve the conclusions section, it is very general and does not clearly explain the main objectives achieved in this research. The conclusions section should present in a clear and summarized way the main parts obtained with this study and the main contributions.
Author Response
Response to Reviewer 4 Comments
Point 4:
The article presents a topic about the Optimization of roasting parameters for recovery of vanadium and tungsten from spent SCR catalyst with composite roasting, however there are points that need to be clarified and summarized.
Abstract: I suggest improving the abstract, making clear the purpose and the main results that have been achieved with this study.
Introduction: The introduction is very general and very short. It should be improved and clearly explain the purpose and final objective of this study with appropriate references. In a research paper, it is expected that introduction section briefly explains the starting background and, even more important, the originality (novelty) and relevancy of the study is well established. Once this is done, hypothesis and objectives of the study need to be addressed, as well as a brief justification of the conducted methodology.
Experimental procedure: Explain in greater detail the experimental processes carried out in the research.
Discussion Section: Create a separate Discussion section. The Discussion section should compare the study by clearly comparing the results obtained by the authors with other studies conducted by other authors.
Conclusions Section: Improve the conclusions section, it is very general and does not clearly explain the main objectives achieved in this research. The conclusions section should present in a clear and summarized way the main parts obtained with this study and the main contributions.
Response 4:
Thank you for taking the time to review my manuscript processes-1427548-org. Based on your review comments, I have made some modifications as following.
- In the part of abstract, I revised the purpose and the main results that achieved with my study in detail.
- In the part of introduction, I added some background explanation about spent SCR catalyst recovery in detail and perfected objectives of the study need to be addressed.
- In the part of materials and methods, I added two pictures about the macroscopic morphology of spent SCR catalyst and explained the spent SCR catalyst used in our study. I added greater detail the experimental processes in the research. At the same time, I added some possible roasting reactions in the part of results and discussion 3.1.
- In the part of results and discussion, (1) I added a XRD diffraction pattern of spent SCR catalyst in 3.1 to explain the minerals phase of spent catalyst. From the XRD diffraction pattern of spent SCR catalyst, I found that the main structure of spent SCR catalyst was anatase TiO2, and owing to the less content and the uniform distribution of V and W, the diffraction peaks of V and W failed to appear in the figure 1. (2) I added a section to compare the melting points of different sodium additives in 3.2.2. From the melting points of different sodium additives in figure 2, it can be seen that NaOH had the lowest melting point, which indicated that adding NaOH into the higher melting point additives can enhance the mass transfer among different materials, thus decreasing the roasting temperature. (3) I added a discussion section to compare with the leaching efficiencies of V and W for other processes obtained by other studies and other authors in 3.7. From the leaching efficiencies of V and W for different processes in table 6, the efficiency of V for the NaOH-NaCl composite roasting was much higher than that for pure soda roasting, and simultaneously the efficiency of W for the NaOH-NaCl composite roasting was the same as that for other sodium roasting methods. Moreover, the roasting temperature of NaOH-NaCl composite roasting cut down by 150 ℃ than that of pure soda roasting. In addition, the efficiency of V and W for the NaOH-NaCl composite roasting (93.25% leaching efficiency of V and 99.17% leaching efficiency of W) were also higher than that for the acid methods (about 85% leaching efficiency of V) or alkali methods (about 92% leaching efficiency of V and 87% leaching efficiency of W). (4) At last, I revised the format, language and inserted a table 5.
- In the part of conclusion, I explained the main results achieved in this research in detail.
Finally, there are still many deficiencies in my manuscript. Owing to being limited by the modification time and scientific research conditions of my university, I failed to finish the XPS characterization, BET characterization and SEM characterization, and I have only corrected most of the deficiencies in this paper. I hope you understand. Thank you for your review!

Round 2
Reviewer 3 Report
can be published. Remove all discursive comments and edit the text again. Bibliographic references are not in the same pattern. Please correct. In the responses, the authors' comments. Source: (Asian) Chinese (China). Does this mean that the result was adapted from another reference? If so, it is necessary to leave in the caption that the result was adapted, having requested authorization from the authors who edited the image.
Cheers,
J. M. De Sousa
Federal Intitute of Educacion, Science and Techonology of Piauí - IFPI
Reviewer 4 Report
Accept in present form